# Determination of Glufosinate-P-Ammonium in Soil Using Precolumn Derivation and Reversed-Phase High-Performance Liquid Chromatography

Lin Chen [1,2,3], Shun Kong [4], Guodong Wang [3], Xiaoju Yan [3], Xuemei Zhang [5], Xiangji Kong [1,2,*] and Yuanqing Bu [1,2]

1   Nanjing Institute of Environmental Science, Ministry of Ecology and Environment of the People's Republic of China, Nanjing 210042, China; wangruij724@gmail.com (L.C.); byq@nies.org (Y.B.)
2   Key Laboratory of Pesticide Environment Assessment and Pollution Control, Ministry of Ecology and Environment of the People's Republic of China, Nanjing 210042, China
3   College of Hydrology and Water Resources, Hohai University, Nanjing 210098, China; qiuzhengzhong1998@gmail.com (G.W.); wshyxj@hhu.edu.cn (X.Y.)
4   Beijing Origin Water Technology Co., Ltd., Beijing 100097, China; kongshun719@gmail.com
5   Jiangsu Institute of Metrology, Nanjing 210023, China; zhangxuemei@jssjlkxyjy.wecom.work
*   Correspondence: kxj@nies.org

**Abstract:** This study developed an analytical method to quantify glufosinate-P-ammonium (GLUF-P) in farmland soil using a reversed-phase high-performance liquid chromatography (HPLC) system with a fluorescence detector after derivatization. GLUF-P in farmland soil was extracted with a mixed alkaline solution and was further derivatized with 9-fluorenyl methyl chloroformate (FMOC) at 25 °C for 1 h. The derivatives were separated with an ACE-C18 column, gradient eluted with a mobile phase A of acetonitrile and a mobile phase B of 0.2% phosphoric acid solution, and finally determined by high-performance liquid chromatography (HPLC) with fluorescence detection at an excitation wavelength of 254 nm and an emission wavelength of 279.8 nm. The limits of detection (LODs) in the four types of soil ranged from 0.004 to 0.015 mg/kg, and the limits of quantification (LOQs) ranged from 0.0125 to 0.05 mg/kg. The mean recoveries of GLUF-P ranged from 94% to 119.8%, and the relative standard deviations (RSDs) varied between 2.8% and 9.0% when the spiked concentrations of GLUF-P were 0.1 mg/kg and 1.0 mg/kg, respectively. The coefficients of regression for the linearity equation were more than 0.99. The proposed method had high sensitivity and could be used for the determination of GLUF-P residues in farmland soil.

**Keywords:** GLUF-P; farmland soil; derivatization; HPLC; fluorescence detection

## 1. Introduction

Glufosinate (GLUF) is a type of herbicide with high efficiency, low toxicity and nonselectivity that was first developed and marketed by the Hoechst Corp [1]. It could effectively affect photosynthesis by inhibiting the synthesis of glutamine and causing the accumulation of ammonium ions in weeds [2,3]. Glufosinate-P-ammonium (GLUF-P), an L-type chiral isomer of GLUF, is obtained by directional synthesis, and the weeding efficacy of GLUF-P is twice that of GLUF [4]. GLUF-P is receiving increasing attention with its marketing. The demand for this type of herbicide is increasing dramatically [5–7], which has posed potential risks to human health and the ecological environment that should not be neglected [8,9]. According to scientific statistics, approximately two-thirds of the active ingredients of GLUF-P enter the soil and water or other environments after its application. Therefore, it is necessary to develop a quick, accurate and highly sensitive determination method for pesticides to objectively evaluate their ecological environmental risk.

GLUF-P is extremely polar and water soluble, and its molecule lacks ultraviolet and fluorescence chromo groups [10], all of which lead to its inability to be detected by gas chromatography alone (GC) [11] or directly by liquid chromatography (LC) equipped with

an array diode detector or fluorescence detector [12]. Currently, gas chromatography with tandem mass spectrometry (GC-MS) has been used for the determination of GLUF after separating it out of water by ion-exchange (IE) columns and derivatization [10,13–15]. GLUF in biological specimens and grains is generally determined by liquid chromatography with mass spectrometry (LC-MS) [16–18] or by LC after derivatization [19]. Zhang et al. [20] derived GLUF and its two metabolites 3-methyl-phosphinico-propionic acid (MPP) and 2-methyl-phosphinico-acetic acid (MPA) with trimethyl orthoacetate (TMOA) in acetic acid, followed by GC with a flame photometric detector (FPD). Alain et al. [13] used the same method to derive GLUF, MPP and MPA and detected the derivatives with GC tandem MS. Yasushi et al. [21] developed a reversed-phase LC method with fluorescence detection to detect GLUF in biological samples after precolumn derivatization with (+)-1-(9-fluorenyl)ethyl chloroformate (FLEC). To our best knowledge, the residual GLUF-P in farmland soil has not been quantified with LC coupled with the fluorescence detection method.

This study developed a method to determine GLUF-P in farmland soil with high-performance liquid chromatography (HPLC), during which an alkaline aqueous solution was used as the extraction solvent on the basis of the solubility of GLUF-P in water, after which GLUF-P in the extract was precolumn-derivatized with 9-fluorenyl methyl chloroformate (FMOC). The derivatives were determined by HPLC with fluorescence detection after purification. The method had the advantages of high sensitivity and a low detection limit.

## 2. Experimental Section

### 2.1. Chemicals and Reagents

GLUF-P at a purity of 98% and FMOC (purity 99%) were purchased from Dr Ehrenstorfer, Augsburg, Germany. Acetonitrile and dichloromethane were of HPLC grade and purchased from Merck, Darmstadt, Germany. Analytically pure phosphoric acid, trisodium phosphate dodecahydrate, trisodium citrate dihydrate, sodium tetraborate decahydrate and hydrochloric acid were obtained from Fisher Scientific, Pittsburgh, USA. HPLC grade pure water (max. conductivity 0.055 μS/cm) was prepared by a Milli-Q water purification system. Analytically pure diatomite, primary secondary amine and Florisil soil were obtained from Fisher Scientific, Pittsburgh, PA, USA. HLB columns were obtained from Waters Scientific, Milford, MA, USA. The glass fiber filters of 0.45 μm were obtained from Whatman, Japan. Atlantis @ T3 column was purchased from Waters Scientific, Massachusetts, USA.

### 2.2. Instruments and Equipment

Waters 2695 HPLC equipped with a Waters 2475 fluorescence detector was from Waters Corp, Milford, MA. High-speed centrifuge 5804 was from Eppendorf Corp, Hamburg, Germany. The vortex meter was from SCILOGEX Corp, Shanghai, China. The KQ-800KDE ultrasonic extractor was from Shumei Corp, Kunshan, China. The E24A Constant temperature digital oscillator was from NBS Corp, USA.

### 2.3. Solution Preparation

Preparation of the standard stock solution of GLUF-P: A standard stock solution of GLUF-P (0.2 mg/mL) was prepared by weighing standard substance powder and then transferring it into a 10 mL brown volumetric flask, followed by dissolving and diluting it with 100 mL of 80% (volume percentage) acetonitrile aqueous solution, and then, the solution was stored in the dark at 4 °C.

Preparation of derivative reagent solutions: (i) Preparation of FMOC aqueous solution (1 mg/mL): 50.0 mg (Net content) of FMOC powder was weighed and transferred into a 50 mL brown volumetric flask, followed by dissolving and diluting it in acetonitrile, and then, it was stored below 4 °C and kept away from light. (ii) Preparation of sodium tetraborate aqueous solution (0.05 mol/L): 1.91 g of sodium tetraborate decahydrate was weighed and dissolved in 100 mL of water.

Preparation of alkaline mixed extraction solution: 11.40 g of trisodium phosphate dodecahydrate and 2.94 g of trisodium citrate dehydrate were weighed, transferred into a volumetric flask, and dissolved and diluted with 1 L of pure water at 0.03 mol/L and 0.01 mol/L, respectively.

### 2.4. Test Soil

The four types of representative soils tested in this study were Jiangxi red soil, Northeast China black soil, Taihu Lake paddy soil, and Shanxi alluvial soil. After natural drying, grinding and screening with a 2 mm sieve, these soils were stored in the dark at 4 °C. The physical and chemical characteristics of the soils are shown in Table 1.

**Table 1.** The basic physicochemical properties of soil.

| Parameters | Soil | | | |
|---|---|---|---|---|
| | **Jiangxi Red Soil** | **Northeast China Black Soil** | **Taihu Lake Paddy Soil** | **Shanxi Alluvial Soil** |
| | Mechanical components | | | |
| Sand (g/kg) | 126 | 21 | 29 | 24 |
| Silt (g/kg) | 343 | 512 | 662 | 546 |
| Clay (g/kg) | 531 | 467 | 309 | 430 |
| pH | 4.39 | 6.58 | 5.04 | 8.38 |
| Organic matter (g/kg) | 8.3 | 28.4 | 20.1 | 11.8 |
| CEC (cmol (+)/kg) | 9.6 | 25.1 | 8.6 | 9.5 |
| Texture | Reddish brown loam | Black loam | Grey silty soil | Yellow-grey silty soil |

### 2.5. Sample Extraction

Twenty grams of the test soil sample was weighed and transferred into a conical flask, and then 50 mL of alkaline mixed extraction solution was added, followed by eddying for 2 min and ultrasonic extraction for 30 min. The extract was transferred into an 80 mL centrifuge tube and centrifuged for 10 min at 10,000 rpm. The supernatant was filtered into a triangular glass flask. Ten milliliters of the filtrate were transferred into another vessel, and an appropriate amount of hydrochloric acid was added to adjust its pH to approximately 9, followed by letting it stand for 10 min.

### 2.6. Purification

The above solution was extracted with dichloromethane to reduce the influence of impurities in soil on the following derivation and determination. That is, 10 mL of dichloromethane was added to the above extract, the mixture was stratified after oscillating extraction, and the underlying organic layer was discarded. The upper aqueous extract was purified again by dichloromethane. Finally, the purified aqueous extract was transferred into a triangular glass flask. Moreover, the other three purifications with 0.1 g of diatomite, Florisil soil or primary secondary amine (PSA), or HLB column were also adopted separately in order to develop the best purification method.

### 2.7. Derivatization

Two milliliters of the purified extract, 0.5 mL of sodium tetraborate solution and 1 mL FMOC were mixed together successively in a 10 mL tube, and the solution was derived for 0.5 h in a shaker at 25 °C. Five milliliters of dichloromethane were added to the mixture and eddied for 2 min to extract derivatized by-products, followed by centrifugation at 10,000 rpm for 5 min. Then, 1 mL of the upper aqueous portion was filtered through a filter for HPLC detection.

### 2.8. HPLC Analysis

A Waters 2695 HPLC equipped with a Waters 2475 fluorescence detector was used for the concentration determination of GLUF-P at an excitation wavelength of 254 nm and an emission wavelength of 279.8 nm. Chromatographic separation was achieved by using an ACE-$C_{18}$ reversed-phase column (25 cm × 4.6 mm inner diameter, 5-μm particle size; ACE

Corp./Luxembourg, England) at a column temperature of 30 °C, and the injection volume was 10 μL. Acetonitrile (eluent A) and 0.2% phosphoric acid aqueous solution (eluent B) were used as mobile phases, and the flow rate was 1.0 mL/min with a total run time of 20 min. The gradient elution procedure is shown in Table 2.

**Table 2.** HPLC gradient elution program at 0, 10, 15 and 20 min.

| Time (Min) | Mobile Phase A (%) | Mobile Phase B (%) |
|---|---|---|
| 0 | 35 | 65 |
| 10 | 25 | 75 |
| 15 | 80 | 20 |
| 20 | 35 | 65 |

*2.9. Method Validation Set*

This method was validated with the linearity of the standard curve, the limit of detection (LOD) and limit of quantification (LOQ), precision and accuracy. Five concentration levels of GLUF-P standard solutions from 0.005 to 0.5 mg/L were prepared by spiking various amounts of the GLUF-P standard stock solution into acetonitrile. Similarly, the matrix standard solution of GLUF-P was prepared by spiking various amounts of the GLUF-P standard stock solution into the mixed alkaline extract, which was prepared from the control soil according to Procedure 2.5. Finally, the prepared concentration of GLUF-P matrix solution ranged from 0.005 to 1 mg/L. Purification and derivation of the GLUF-P standard solution series or the matrix standard solution series were performed according to Procedures 2.6 and 2.7, respectively. Finally, the derivatized solution was detected under the conditions described in Section 2.8. Linear calibration curves were obtained by plotting the peak areas (y) and concentrations of GLUF-P (x), and the correlation coefficient (r) of the standard calibration or matrix-matched calibration curve was determined. Selectivity was assessed by analyses of three matrix blanks for evaluation against solvent blanks. The accuracy and precision of the method represented by the recoveries and relative standard deviations (RSDs) separately were evaluated by GLUF-P recovery experiments from four types of soils separately at two spiked levels of 0.1 or 1 mg/kg with five replicates. The LOD was assessed by a signal-to-noise (S/N) ratio of 3. The LOQ was $3.3 \times$ LOD

## 3. Results and Discussion

*3.1. Optimization of Derivatization Conditions*

The determination of GLUF-P in samples without derivatization was adopted in some studies [18]; however, the levels of testing conditions were high. The chromatographic columns should have good retention performance to avoid the test substance being eluted too fast to be detected, and MS detectors in the determination systems were also necessary. However, the shortcomings of such a method include an obvious matrix effect, poor peak shape and column damage [22]. GLUF-P in the complex matrix of the soil sample was determined by HPLC with fluorescence detection after derivatization in our study referring to available data, in which results with high sensitivity without matrix effects were expected to be attained. In this section, the effects of the derivatization temperature and period on the determination are mainly discussed: the blank quartz sands that replaced the soil were pretreated according to the methods in Sections 2.5 and 2.6, and then, the extract obtained was used as a solvent to prepare the matrix working solutions at 0.005~0.5 mg/L. The matrix working solutions were subsequently derivatized at 25 °C and 40 °C separately according to the methods in Section 2.7 and were finally determined by HPLC under the conditions of Section 2.8, and the working curves at different temperatures were plotted separately. The measured working curves are shown in Figure 1 by the end of the 1 h derivation time. The linear correlation coefficient r of the working curve at 25 °C was higher than 0.999. The corresponding peak area values for the same concentrations of working solutions at 40 °C were significantly lower than those at 25 °C. The chromatographic response to the derivatives at 40 °C was approximately 40% of that at 25 °C. When the

concentration of GLUF-P was 0.005 mg/L, the derivatization effect at 40 °C was less unsatisfactory. Therefore, 25 °C was the optimum derivatization temperature of GLUF-P.

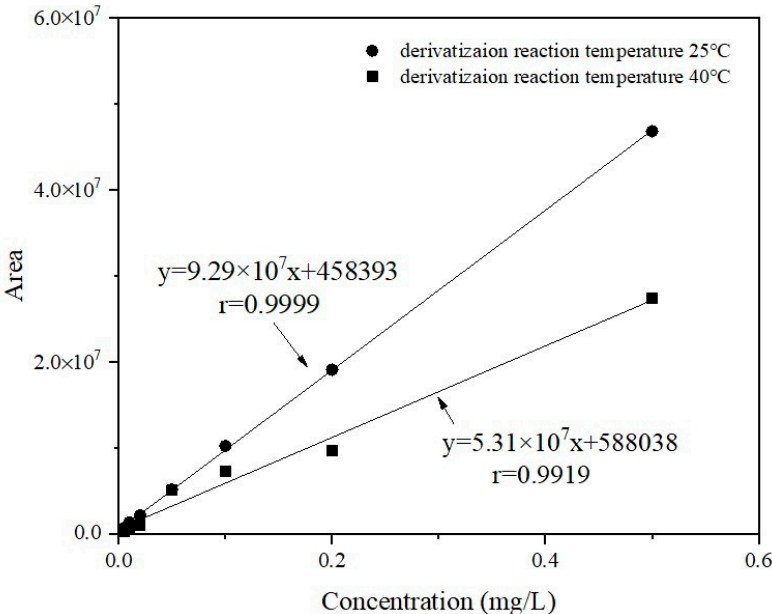

**Figure 1.** The working curves of GLUF-P at 25 °C and 40 °C.

All three series of GLUF-P standard solutions from 0.005–0.5 mg/L were derivatized by different reaction times to attain the optimum reaction period. The results are shown in Figure 2. The peak areas of the derivatives corresponding to the same concentration of GLUF-P were similar at 1 h, 2 h, or 3 h, which suggested that the concentrations of the derivatives did not increase significantly after 1 h. Therefore, 1 h was the optimum reaction period. The chromatograms of the derivatives of the control and 0.1 mg/L GLUF-P standard solutions are shown in Figure 3a,b, respectively, and the retention time of GLUF-P derivatives was within 15–17 min.

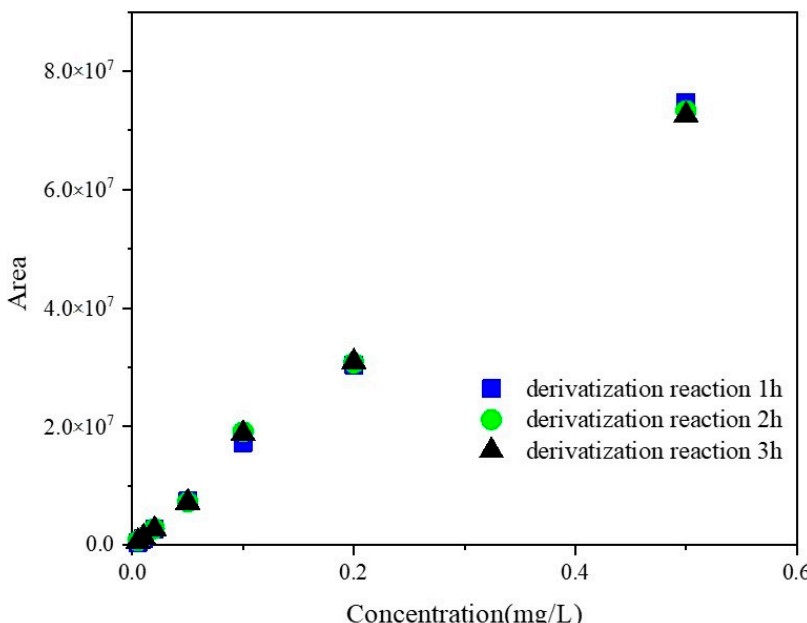

**Figure 2.** Peak areas of derivatives from 0.005 and 0.5 mg/L GLUF-P at 1 h, 2 h, and 3 h.

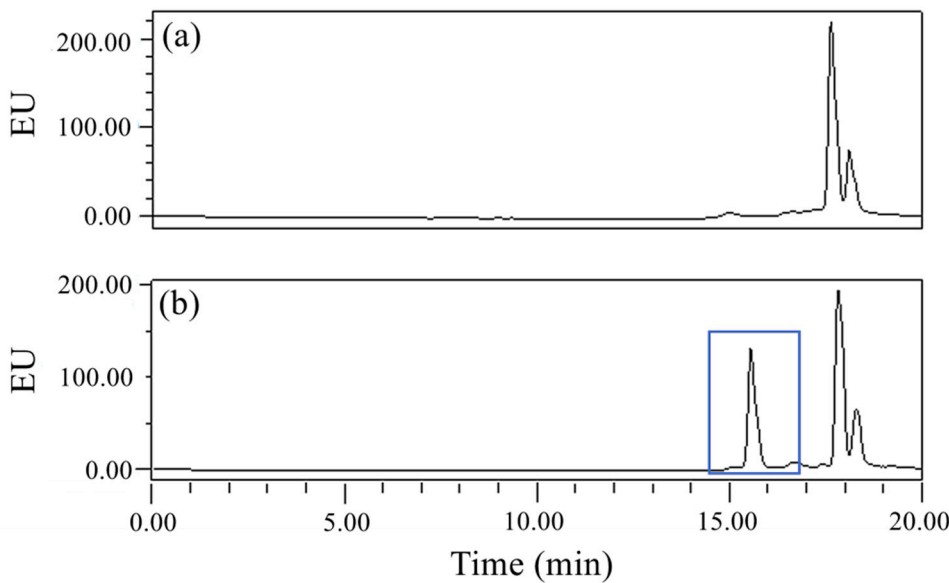

**Figure 3.** Chromatograms for (**a**) samples derived from the control sample and (**b**) 0.1 mg/L standard solution of GLUF-P.

### 3.2. Comparison of Purification Effects

The purification effect experiments were designed at one GLUF-P concentration in four types of farmland soil with four types of purifying agents, in which three parallel samples were needed. A standard solution of GLUF-P was added to each 20 g of soil to prepare a 1.0 mg/kg sample, and then, ultrasonic extraction, centrifugation and filtration were performed sequentially according to Section 2.5. Three milliliters of all of the filtrates were transferred into centrifuge tubes, and 0.1 g of diatomite, Florisil soil or primary secondary amine (PSA) was added separately. Two milliliters of supernatant were taken for derivatization and determination after oscillation and centrifugation. The HLB columns were activated by ultrapure water and the alkaline mixture successively, 3 mL of the filtrate was separated through the columns, and 2 mL of the percolate was collected. All purified solutions were derivatized according to Section 2.7 for 1 h. One milliliter of the supernatant was pushed through the membrane for detection after the derivatized solution was extracted with dichloromethane and centrifuged.

The chromatographic peaks of the GLUF-P derivative could not be found after purification by PSA, mostly because of its adsorptivity. The chromatogram of the derivative from the red soil sample is shown in Figure 4, in which it was demonstrated that PSA was not suitable for the purification of GLUF-P in the soil.

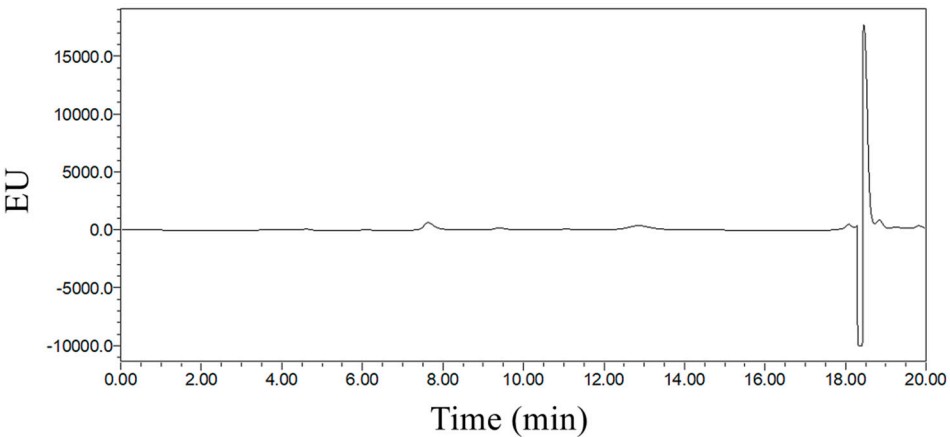

**Figure 4.** Purification effect of PSA on GLUF-P determination in Jiangxi red soil samples.

The recoveries in purification experiments in four soils by diatomite, Florisil soil and HLB columns are shown separately (Figure 5). As shown in Figure 5, the average recovery in Northeast black soil purified by diatomite was 106%, and all of the other recoveries in the other three soils were over 120%. Moreover, the RSDs from the Shanxi alluvial soil group were more than 20%, and the recoveries were 125%, 105% and 160%, respectively. The average recoveries in the four soils purified by Florisil soil and HLB columns were all over 120%. Therefore, the performance of the above purifying agents could hardly meet the quality control requirements. It was concluded that the above purifying agents could not completely remove impurities in the soil extracts, which led to obvious interferences from impurities in the derivatives with the chromatographic peaks, and finally, it was difficult to accurately determine the concentrations of the derivatives due to their poor peak widths and shapes.

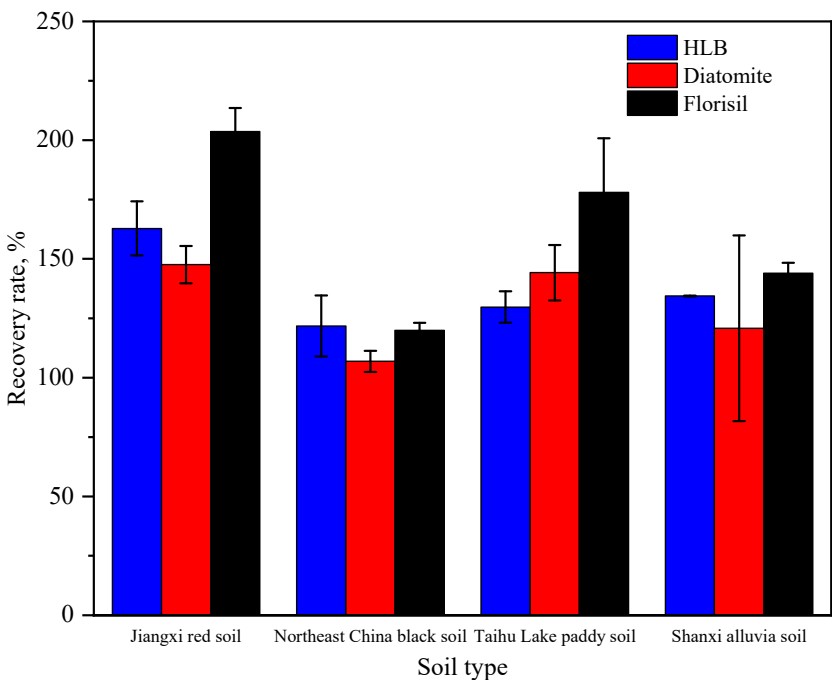

**Figure 5.** Recoveries of GLUF-Ps derived by different purifications in four soils.

Therefore, the following methods were adopted to improve the purification effect. We intensified the centrifuge operation parameters, including increasing the speed to 10,000 rpm and the operation time to 10 min to fully remove the insoluble complex from the alkaline extracts. In addition, liquid-liquid extraction with dichloromethane was adopted to remove soluble impurity derivative by-product interference.

### 3.3. Choice of the Chromatographic Column

Chromatographic separation performances of GLUF-P derivatives by three common chromatographic columns (ACE-$C_{18}$, $T_3$ and RP-$C_{18}$) were compared. The blank control, 0.1 mg/L and 0.5 mg/L GLUF-P matrix solutions that were prepared from Shanxi alluvial soil were determined with three chromatographic columns after derivation. The chromatograms of the ACE-$C_{18}$, $T_3$ and RP-$C_{18}$ columns are shown in Figures 6–8, respectively.

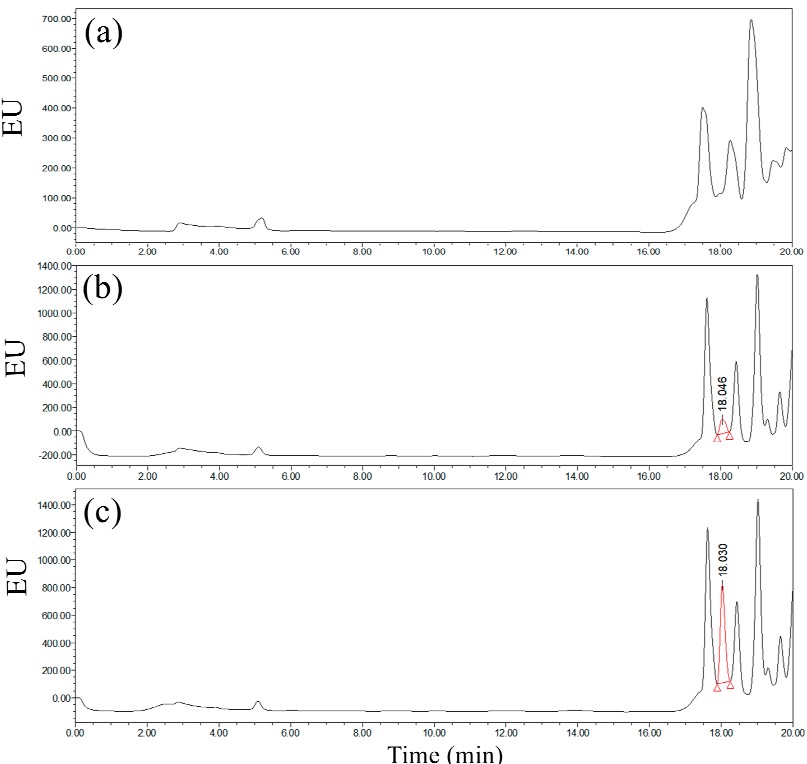

**Figure 6.** Chromatograms on the ACE-C$_{18}$ column for the (**a**) sample derived from the blank control, (**b**) sample derived from the 0.1 mg/L GLUF-P matrix solution, and (**c**) sample derived from the 0.5 mg/L GLUF-P matrix solution.

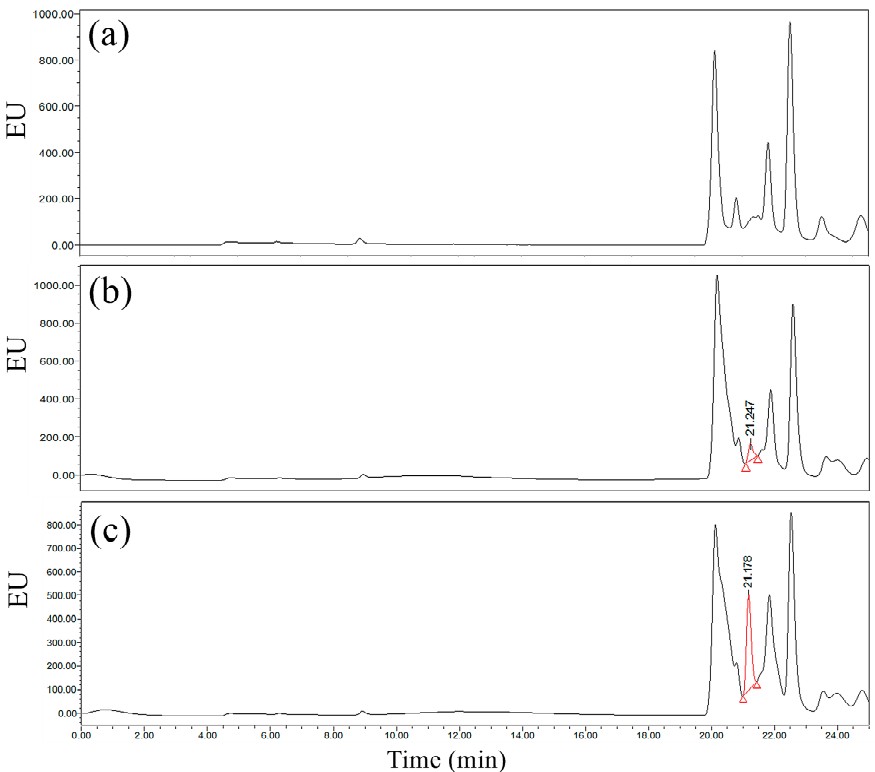

**Figure 7.** Chromatograms on the T$_3$ column for the (**a**) sample derived from blank control, (**b**) sample derived from the 0.1 mg/L GLUF-P matrix solution, and (**c**) sample derived from the 0.5 mg/L GLUF-P matrix solution.

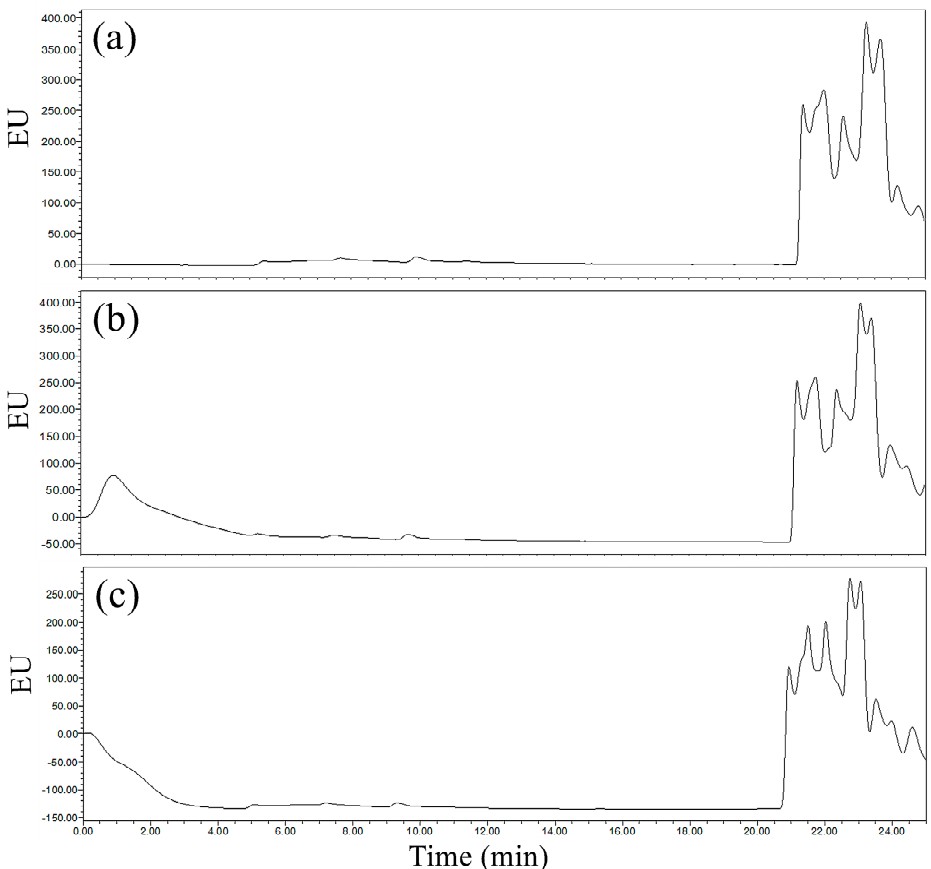

**Figure 8.** Chromatograms of the RP-$C_{18}$ column for the (**a**) sample derived from the blank control, (**b**) sample derived from the 0.1 mg/L GLUF-P matrix solution, and (**c**) sample derived from the 0.5 mg/L GLUF-P matrix solution.

Due to the high water solubility of GLUF-P derivatives, a mobile phase consisting of a high proportion of water was needed to attain sufficient retention on the chromatographic column. Specifically, the proportion of phosphoric acid solution as mobile phase B was between 20% and 80%, and if the samples were separated by a $C_{18}$ common column, the retention time of the target compounds became increasingly shorter with worse separation efficiencies between the chromatographic peaks, which was called 'stationary phase collapse' or 'hole dewetting'. In addition, the peak area of the target substance would be reduced because of contact area loss. As shown in Figure 6, an ACE-$C_{18}$ column was used to separate the derivatized products from the control, 0.1 mg/L and 0.5 mg/L GLUF-P, and the chromatographic retention time was 18.0~18.1 min. The target substance could be separated from other impurities well, and the chromatographic peak shape was also good, which was directly related to the characteristics of the ACE-$C_{18}$ column: the complete phenyl functional group in the stationary phase of the column could prevent 'hole dewetting' or subsequent retention time loss. The stationary phase of the ACE-$C_{18}$ column was composed of ultra-inert and high-purity silica gel, which could maintain the target compound in a good hydrophilic group peak shape.

Figure 7 shows that the derivatives from the standard samples of 0.1 mg/L and 0.5 mg/L GLUF-P had good responses on the chromatograms after separation by the $T_3$ column, which was related to the characteristics of the $T_3$ column. The $T_3$ column was called a 'special column for polarity analysis', whose stationary phase could be compatible with the mobile phase of 100% aqueous solution and had a strong reverse retention capability for the polar analytes. The chromatographic retention time of the derivative of GLUF-P by the $T_3$ column was between 21.1 and 21.2 min. Therefore, the chromatographic retention time by the $T_3$ column was approximately 3 min longer than that by the ACE-$C_{18}$ column

under the same test conditions, and the peak symmetry was worse than the latter, which could result in integration error and quantification accuracy.

No significant absorption peaks of derivatives were found for the samples of 0.1 mg/L and 0.5 mg/L GLUF-P by the RP column during 0~26 min (Figure 8). It was concluded that there were two extreme possibilities: One was that the derivatives had no retention on the RP column and another was that the retention times of the targets on the RP-C$_{18}$ column exceeded 26 min. However, the RP-C$_{18}$ column had good analytical selectivity and could provide good retention for polar materials due to its polarity chimerism and hydrophobic chain structure, which led to the retention times of GLUF-P derivatives being longer than 26 min, thus, it was not suitable for the batch test due to its worse efficiency.

Comprehensively, it was indicated that the ACE-C$_{18}$ column was the most suitable column for the determination of GLUF-P derivatives for better efficiency and response.

### 3.4. Method Validation Results

The concentration-peak area standard curve for GLUF-P over the range of 0.005 to 0.5 mg/L was drawn after the determination. The linear regression equation $y = 1.42 \times 10^8 x - 6545$ with a correlation coefficient r above 0.999 was obtained

The matrix working curve equations and linear ranges of the four soils are shown in Table 3. Moreover, the LOD in different soil was relevant to the physicochemical properties of soil, such as the pH, and mechanical components, which maybe generate interference with the retention time and the basic line, even for the peak area.

**Table 3.** The linear ranges, equations and correlation coefficients (r), LODs and LOQs of GLUF-P in soils.

| Index | Jiangxi Red Soil | Northeast China Black Soil | Taihu Lake Paddy Soil | Shanxi Alluvial Soil |
|---|---|---|---|---|
| Linear range (mg/L) | 0.02~1.0 | 0.01~0.5 | 0.005~0.5 | 0.01~0.5 |
| Equation | $y = 1.0 \times 10^8 x - 2.0 \times 10^6$ | $y = 8.0 \times 10^7 x + 489,600$ | $y = 3 \times 10^7 x + 1,000,000$ | $y = 9 \times 10^7 x - 50,952$ |
| r | 0.998 | 0.999 | 0.992 | 0.999 |
| LOD (mg/kg) | 0.015 | 0.008 | 0.004 | 0.008 |
| LOQ (mg/kg) | 0.050 | 0.025 | 0.0125 | 0.025 |

The recoveries and the precision of GLUF-P determination in the four soils are shown in Table 4. The average recoveries at the 0.1 and 1.0 mg/kg concentrations of GLUF-P in the four soils ranged from 94% to 119.8%, with RSDs of 2.8% to 9.0%.

**Table 4.** Recovery (%) and RSD (%) data of GLUF-P (n = 5).

| Item | Jiangxi Red Soil | Northeast China Black Soil | Taihu Lake Paddy Soil | Shanxi Alluvial Soil |
|---|---|---|---|---|
| Spiked level (mg/kg) | | | 0.1 | |
| Recoveries (%) | 116.4~120.3 | 106.8~115.4 | 91.4~96.8 | 106.5~114.2 |
| Average Recoveries | 119.8 | 111.7 | 94.0 | 110.2 |
| RSD(%) | 3.0 | 6.3 | 8.5 | 2.8 |
| Spiked level (mg/kg) | | | 1.0 | |
| Recoveries (%) | 112.2~118.3 | 115.5~118.4 | 109.6~116.2 | 112.4~114.6 |
| Average Recoveries (%) | 116.6 | 116.7 | 112.1 | 113.2 |
| RSD (%) | 3.3 | 8.4 | 9.0 | 3.8 |

The LOD was defined at the minimum concentration of the matrix standard solution that could be detected reliably and was converted according to the signal-to-noise ratio of 3 (S/N). Therefore, the LODs of GLUF-P in the four soils were 0.015 mg/kg in Jiangxi red soil, 0.008 mg/kg in Northeast China black soil, 0.004 mg/kg in Taihu paddy soil and 0.008 mg/kg in Shanxi alluvial soil. The LOQs were the minimum quantified concentration with recoveries of 70~120%, and the RSDs were less than 20%. Therefore, the LOQs in the four soils were 0.05 mg/kg in Jiangxi red soil, 0.025 mg/kg in Northeast China black soil, 0.0125 mg/kg in Taihu paddy soil and 0.025 mg/kg in Shanxi alluvial soil.

The determination performance comparison among existing research was shown in Table 5. It was concluded that the LOD of the paper was almost the lowest among all studies. Compared with other studies, the recoveries were relatively reasonable.

**Table 5.** Determination performance data of GLUF-P in different matrix by different method.

| Method | Matrix | LOD | Recoveries, % | References |
|---|---|---|---|---|
| Derivation and HPLC | Maize | 0.02 mg/kg | 98.0–100.5 | [6] |
| Derivation and HPLC | human serum | 0.005 mg/L | 95.1–98.7 | [21] |
| Derivation and GC-MS | human serum | 0.1 mg/L | 38.8–41.6 | [11] |
| Derivation and GC-MS | water | 0.01 mg/L | 71.0–125.0 | [13] |
| SPE -LC-MS | soybean and corn | 0.014mg/kg | 92.0–104.0 | [18] |
| SPE- UPLC-MS | Soil | 0.02 mg/kg | 80.0–91.6 | [23] |
| Derivation and HPLC | Soil | 0.004–0.015 mg/kg | 94.0–119.8 | This study |

## 4. Conclusions

In our study, a HPLC with a fluorescence determination method for GLUF-P in soil was developed. The preferred conditions for the analysis of GLUF-P in four representative soils in China were ascertained after optimizing the derivation conditions and comparing different purifying agents and chromatographic columns. It was suggested that this method can be applied to determine the GLUF-P in farmland soil because of satisfactory validation parameters for linearity, high precision, and low detection limits.

**Author Contributions:** Data curation L.C.; original draft preparation, S.K.; Writing review & editing, G.W. and X.K.; methodology, X.Y.; validation, X.Z.; Formal analysis, X.K.; resources, Y.B. All authors have read and agreed to the published version of the manuscript.

**Funding:** The research was funded by the Central Scientific Research Projects for Public Welfare Research Institutes grant number [GYZX210301].

**Institutional Review Board Statement:** Not applicable.

**Informed Consent Statement:** Not applicable.

**Acknowledgments:** The authors appreciated the Central Scientific Research Projects for Public Welfare Research Institutes (Grant No.: GYZX210301).

**Conflicts of Interest:** The authors declare no conflict of interest.

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
