# Peer review of "Determination of Glufosinate-P-Ammonium in Soil Using Precolumn Derivation and Reversed-Phase High-Performance Liquid Chromatography"

_water, doi:10.3390/w14111816_

Round 1

Reviewer 1 Report

The manuscript deals with the analysis of a widespread potentially harmful herbicide with simple chromatographic equipment (no mass spec) and easy sample prep. Thus the manuscript is of interest. However, there are some shortcomings and issues that I would like to be addressed before I can recommend the paper for publication.

There are some gramatical mistakes and strange choice of wording like referening to the sample: "standing for 10 mins" and "one was no retention..." I would suggest to proof read the MS once more. I am no native English speaker either so I know the struggles of writing a paper in English but unforutantely bad grammar can severly influence the perceived quality of an MS. Also: CH2Cl2 is named both as dichloromethane and methylene chloride. Please stick to one name throughout the MS.

Placing the experimental section at the end of the MS is really odd and makes it hard to follow the paper. I know there are no strict style guideline at the journal but this just ruins the reading experience. Please put the experimental section between the introduction and results as every other paper would do.

Several equipment used during the method development is missing from the equipment section (the T3 column, the HLB SPE cartridge, Florisil... etc.). The parameters evaluated during the method development should be present in the experimental section not just in the results. Right now the only place I can find what paramters were evaluated are in the results. Add these info to the experimental section as well. The type of filter is also missing from this section. Was it a paper filter, a membrane syringe filter? What was the material of it? Also, the injection volume is missing from the MS, please add the info.

Is Table 4 valid? Does the gradient really goes (for A%): 35->25->80->35? That makes no sense for me. Why would you lower the ACN content then increase it back during the run? The end part (80->35) I understand, it is for equlibration, but what is with the 35>25>80 part? Please explain this.

Does this method really robust? Seems like everything comes at the end of the chromatogram and you have to get the target compound from a sea of peaks. I would have made a gradient which changes fast at the start where nothing of interest is coming and slow down considerably to let compounds separate where it is important. How do you guarantee that the GLUT-P peak is always integrated without error from the other peaks next to it? Also: you wrote that the other C18 column, which you did not choose at the end had better selectivity and separation capabilites but it possibly took longer to elute GLUT-P. Why is this a problem? If you have a long ret time you could just modify the gradient to suit the column. Why did you stick to this one defined gradient and discard the better column and not the other way around while you desperately needed separating power because of the matrix?

How does your method stand up against the cited one by Yasushi et al.? Seems like the basics of the two methods are the same. Your method is better in which ways? Faster, cheaper, better LOD/LOQ, selectivity... etc?

Fig. 1 is terribly pixelated, please add it in a higher quality. Also, I would like to see a fig. with the chromatograms of each soil sample spiked to the LOQ value with the GLUT-P integration highlighted just like on Fig 6 and 7.

These issues have to be sorted out before the manuscript could be accepted. Therefore I request a revision to the MS at this time.

Reviewer 2 Report

Manuscript ID water-1681891 Title:Determination of glufosinate-P-ammonium in farmland soil using precolumn derivation with 9-fluorenyl methyl chloroformate and reversed-phase high-performance liquid chromatography.   Please do the following:

This study developed an analytical method to quantify glufosinate-P-ammonium (GLUF-P) in farmland soil using a reversed-phase high-performance liquid chromatography (HPLC) system with a fluorescence detector after derivatization. GLUF-P in farmland soil was extracted with a mixed alkaline solution, and the extract was derivatized with 9-fluorenyl methyl chloroformate (FMOC) at 25℃ for 1 h.

The authors explained the study very well and to make it much better the following points must be added to the manuscript;

  • The title is too long and it can be shortened with more informative.
  • A table should be added in the results section showing a comparison between at least 5 studies of measuring Glufosinate and showing LOD and their main recoveries with references.
  • How do you compare your method using the HPLC and derivatization which consume a lot of time and chemicals with using LC/MS/MS.
  • Please mention something about the effect of the physicochemical properties of soil on using the method.
  • Drop the plagiarism in the conclusion section.

Reviewer 3 Report

I have some questions for this work.

The chromatogram of this work is not beautiful, I think the author should be  adjusted the mobile phase composition to speed up the analyte peak. 

Why the author used HPLC column of 25 cm, I think 15 cm is enough.

Why did the author spike over the linear range?

Round 2

Reviewer 1 Report

The paper has been considerably improved and the issues I highlighted previously have been resolved. Based on the answers by the authors I now understand why the gradient is odd. I would have modified it to be like a typical RP-LC gradient, but I can accept your reasoning. In its current state, the manuscript can be accepted.

Reviewer 3 Report

-